# Pulmonary sarcoidosis with and without extrapulmonary involvement: a cross-sectional and observational study in China

Cheng-Wei Li,[1] Ru-Jia Tao,[2] Dan-Feng Zou,[3] Man-Hui Li,[2] Xin Xu,[2] Wei-Jun Cao[2]

C-WL, R-JT and D-FZ contributed equally.

¹Department of Respiratory and Critical Care Medicine, Shanghai Pulmonary Hospital, Soochow University, Suzhou, China
²Department of Respiratory and Critical Care Medicine, Shanghai Pulmonary Hospital, Tongji University School of Medicine, Shanghai, China
³Clinic and Research Center of Tuberculosis, Shanghai Key Lab of Tuberculosis, Shanghai Pulmonary Hospital, Tongji University School of Medicine, Shanghai, China

**Correspondence to**
Professor Wei-Jun Cao;
weijuncao@126.com

## ABSTRACT

**Objectives** Sarcoidosis is a multisystem disease characterised by the formation of granulomas within various organs, mainly the lungs. Several studies from different countries have been undertaken to investigate sarcoidosis with extrapulmonary involvement except from China. The objective of this study is to investigate a comparative clinical analysis in patients with pulmonary sarcoidosis with and without extrapulmonary involvement from China.

**Methods** Data from inpatients diagnosed with sarcoidosis at Shanghai Pulmonary Hospital (Shanghai, China) between January 2009 and December 2014 were retrospectively collected and analysed. Six hundred and thirty-six patients with biopsy-proven sarcoidosis were included in the study, including 378 isolated pulmonary sarcoidosis and 258 pulmonary sarcoidosis plus extrapulmonary involvement.

**Results** Two hundred and fifty-eight (40.6%) patients with pulmonary sarcoidosis had extrapulmonary involvement. Extrapulmonary localisations were detected mostly in extrathoracic lymph nodes (n=147) and skin (n=86). Statistically significant differences were demonstrated between patients with pulmonary sarcoidosis plus extrapulmonary involvement and patients with isolated pulmonary sarcoidosis for fatigue (16.6%vs8.3%, P<0.05), serum ACE (SACE) levels (79.0±46.9 IU/L vs 69.7±38.7 IU/L, P<0.05), and high-resolution CT (HRCT) findings (53.8%vs46.2%, P<0.05).

**Conclusions** Extrapulmonary involvement is common in patients with pulmonary sarcoidosis, with the most common sites being extrathoracic lymph nodes and skin. Patients with sarcoidosis with extrapulmonary involvement are more symptomatic (fatigue), have higher SACE levels and more deteriorating HRCT findings, to which clinicians should pay attention.

## Strengths and limitations of this study

► The present study provides a large number of patients with pulmonary sarcoidosis with and without extrapulmonary involvement from China.
► The patients with sarcoidosis had a long-term follow-up (24–42 months) on chest high-resolution CT.
► There is a risk of selection bias due to the retrospective design.

## INTRODUCTION

Sarcoidosis is a granulomatous disease of unknown aetiology. It is characterised by the formation of immune granulomas in many involved organs.[1 2] Sarcoidosis is a worldwide disease, with its prevalence ranging from 4.7 to 64 in 100 000, and its incidence ranging from 1.0 to 35.5 in 100 000 per year.[3] However, limited epidemiological data in China are available.

Sarcoidosis is a multisystem disease. Although the lungs are most commonly involved, pulmonary sarcoidosis is often accompanied by the involvement of extrapulmonary organs,[1 4] such as the skin, eyes and heart that are significantly affected.[5] A Case Control Etiologic Study of Sarcoidosis (ACCESS) showed that among 736 patients with sarcoidosis, 95% of the patients had pulmonary involvement, and 368 out of these 736 patients (50%) had concomitant extrapulmonary involvement.[6] Extrapulmonary involvement can be the major presentation of the disease and sometimes may be life-threatening.[5 7] Moreover, the assessment of pulmonary and extrapulmonary organ involvement is important for clinical treatment decisions.[8]

It is well known that the features of sarcoidosis have varied widely among various populations around the world. To date, several studies from different countries have been undertaken to investigate sarcoidosis with extrapulmonary involvement. However, there is limited information in Chinese patients with sarcoidosis. Therefore, the purpose of this study is to compare the clinical characteristics and prognosis of patients with pulmonary sarcoidosis with and without extrapulmonary involvement from China.

## METHODS

Data were retrospectively collected from inpatients diagnosed with sarcoidosis at the Shanghai Pulmonary Hospital (Shanghai, China) between January 2009 and December 2014. Patients who had not received a chest radiograph or chest high-resolution CT (HRCT) scan examination or who had not received tissue biopsy were excluded. All aspects of the study were performed in accordance with relevant guidelines and regulations.

Diagnosis of sarcoidosis is based on standardised criteria,[1] which detailed the features of sarcoidosis, including clinical and radiological presentation, and pathological findings (non-caseating epithelioid granulomas), as well as evidence of no alternative diseases (infections, particularly tuberculosis; occupationally induced, environmentally induced and drug-induced granulomatosis; common variable immune deficiency; Blau's syndrome; sarcoid-like reactions in cancers and lymphomas, and other idiopathic granulomatosis).[3]

Organ involvement was determined and classified in each patient in accordance with the criteria proposed in the ACCESS formulation,[9] based on clinical assessment and widely available tests, including the evolution of ECG, positron emission tomography scans and MRI that can improve the detection of the disease. According to the organs affected by sarcoidosis, patients were divided into two groups: isolated pulmonary sarcoidosis and pulmonary sarcoidosis with extrapulmonary involvement.

The following variables were obtained from the medical records: general and anthropometric information, symptoms, chest radiographs or CT films, serological indicators and pulmonary function parameters. The chest radiographs or CT films were read at each site by two investigators; the findings were evaluated and staged according to the modified Scadding criteria.[10] Pulmonary function studies were performed and a percentage of the subject's predicted value was calculated,[11] airflow limitation was defined as a forced expiratory volume in 1 s/forced vital capacity (FEV1/FVC) ratio below the fifth percentile of the predicted value, and the restrictive pattern was defined as FVC below the predicted lower limit, with a normal FEV1/FVC ratio. The patients were followed up on chest HRCT at the outpatient clinic of Shanghai Pulmonary Hospital. The HRCT results were assessed by two independent radiologists. The deteriorating chest HRCT findings (an increase in extent compared with the previous HRCT results) were recorded in patients after their discharge from the hospital (figure 1). Written informed consent was obtained from all of the patients. All aspects of the study were performed in accordance with relevant guidelines and regulations.

### Statistical Analysis

Quantitative variables were expressed as the mean±SD, and qualitative variables were expressed as absolute numbers and percentages. The Kolmogorov-Smirnov test

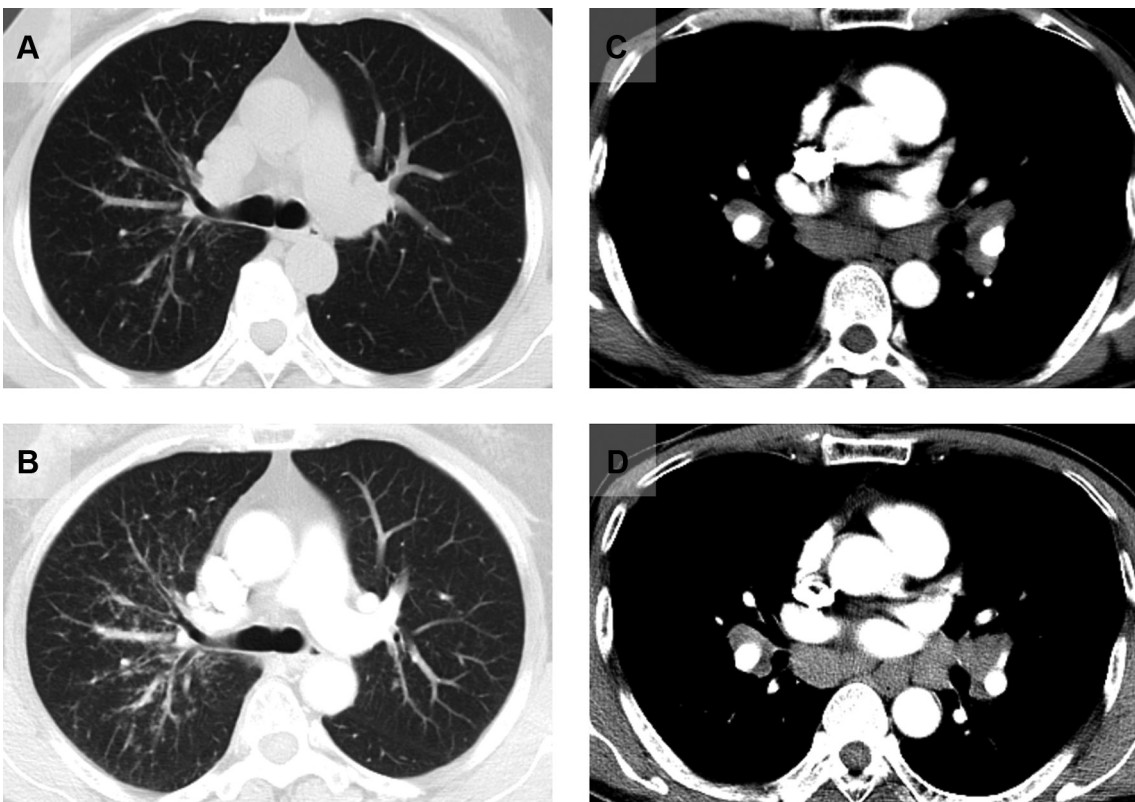

**Figure 1**  High-resolution CT findings in patients with sarcoidosis. (A) Pulmonary micronodules at the time of initial hospitalisation. (B) Pulmonary involvement during follow-up. (C) Enlarged mediastinal lymph node at the time of initial hospitalisation. (D) Larger mediastinal lymph node during follow-up.

**Table 1** Organ involvement in patients with sarcoidosis

| Organ | n (%) |
|---|---|
| Isolated pulmonary involvement | 378 (59.4) |
| Pulmonary sarcoidosis with extrapulmonary involvement | 258 (40.6) |
| Extrathoracic lymph node | 147 (23.1) |
| Skin | 86 (13.5) |
| Eyes | 8 (1.3) |
| Bone/joints | 8 (1.3) |
| Liver | 4 (0.6) |
| Nose | 4 (0.6) |
| Parotid/salivary | 3 (0.5) |
| Spleen | 2 (0.3) |
| Renal | 2 (0.3) |
| Cardiac | 1 (0.2) |
| Muscles | 1 (0.2) |

Total numbers add up to greater than the total number of patients because eight patients had more than one organ involvement.

was used to analyse the distribution of variables. In the bivariate analysis, the t-test for independent variables was used to analyse normally distributed variables, and the Mann-Whitney U-test was used to analyse non-normally distributed variables. Qualitative variables were compared using the $\chi^2$ test. A logistic regression model was used to determine the factors that were independently associated with deteriorating chest HRCT findings. The variables that presented statistically significant differences ($P<0.05$) in the bivariate analysis and were of clinical interest were included as the independent variables in the model. The OR and 95% CI were calculated for the independent variables. Statistical analysis was performed using SPSS V.22.0 package software with a value of $P<0.05$ considered statistically significant, and graphs were drawn using GraphPad Prism V.5 software.

## RESULTS

A total of 636 patients with biopsy-proven sarcoidosis (378 isolated pulmonary sarcoidosis and 258 pulmonary sarcoidosis with extrapulmonary involvement) were included in the study. Among these 636 patients, 298 responded fully to the follow-up study on HRCT (182 isolated pulmonary sarcoidosis and 116 pulmonary sarcoidosis with extrapulmonary involvement), the median follow-up time after discharge was 30 months (ranging from 24 to 42 months). Thirty-nine patients had deteriorating chest HRCT findings after their discharge from the hospital.

The organ involvement and baseline characteristics of the subjects are shown in tables 1 and 2, respectively; the serological and pulmonary function parameters are shown in tables 3 and 4, respectively. Extrapulmonary localisations were detected mostly in extrathoracic lymph nodes (n=147) and skin (n=86). There were statistical

**Table 2** Baseline and clinical characteristics of subjects with pulmonary sarcoidosis, with and without extrapulmonary involvement

| Parameter | Isolated pulmonary sarcoidosis | Pulmonary sarcoidosis with extrapulmonary involvement | P value |
|---|---|---|---|
| Subjects, n (%) | 378 (59.4) | 258 (40.6) | – |
| Sex, M/F, n | 260/118 | 174/84 | 0.721 |
| Age, years | 48.8±10.7 | 48.2±11.5 | 0.613 |
| BMI, kg/m$^2$ | 23.9±3.3 | 23.4±3.3 | 0.033 |
| Smoking history, n (%) | 59 (15.6) | 29 (11.2) | 0.114 |
| Occupational exposure, n (%) | 33 (8.7) | 18 (7.0) | 0.424 |
| Symptoms, n (%) | 277 (73.3) | 181 (70.2) | 0.467 |
| Cough, n (%) | 231 (83.4) | 142 (78.5) | 0.127 |
| Chest congestion, n (%) | 80 (28.9) | 50 (27.6) | 0.548 |
| Dyspnoea, n (%) | 50 (18.1) | 36 (19.9) | 0.793 |
| Chest pain, n (%) | 39 (14.1) | 19 (10.5) | 0.204 |
| Fatigue, n (%) | 23 (8.3%) | 30 (16.6%) | 0.013 |
| Fever, n (%) | 18 (6.5) | 10 (5.5) | 0.593 |
| Scadding stage | | | |
| Stage 1, n (%) | 128 (33.9) | 67 (25.9) | 0.034 |
| Stage 2, n (%) | 232 (61.4) | 180 (69.8) | 0.0296 |
| Stage 3, n (%) | 18 (4.7) | 11 (4.3) | 0.7674 |

Data are presented as mean±SD or %.
BMI, body mass index; F, female; M, male.

**Table 3** Serological indicators of subjects with pulmonary sarcoidosis, with and without extrapulmonary involvement

| Parameter | Isolated pulmonary sarcoidosis | Pulmonary sarcoidosis with extrapulmonary involvement | P value |
|---|---|---|---|
| CRP, mg/L | 4.0±4.2 | 3.8±3.0 | 0.547 |
| TB-AB, n % | 22 (5.8) | 20 (7.8) | 0.335 |
| sIL-2R, U/mL | 1151±842 | 1163±798 | 0.32 |
| UC, mmol/24 hours | 6.09±2.94 | 6.08±2.65 | 0.117 |
| SC, mmol/L | 2.39±0.13 | 2.42±0.15 | 0.022 |
| SACE, IU/L | 69.7±38.7 | 79.0±46.9 | 0.016 |
| ALP, IU/L | 86.4±33.4 | 84.7±33.4 | 0.259 |
| ALB, g/L | 40.16±4.02 | 40.81±3.86 | 0.8 |
| GLB, g/L | 27.96±5.34 | 30.14±5.67 | 0.013 |
| A/G | 1.47±0.28 | 1.42±0.27 | 0.064 |
| UA, µmol/L | 309.4±85.4 | 308.6±99.3 | 0.728 |
| Seroglobulin β | 11.23±1.71 | 11.30±1.43 | 0.278 |
| Seroglobulin γ | 17.72±3.93 | 18.40±4.13 | 0.084 |
| WCC, $10^9$/L | 6.2±2.1 | 5.7±2.0 | 0.377 |
| ESR, mm/hour | 15.8±10.7 | 15.8±9.3 | 0.288 |

Data are presented as mean±SD or %.
A/G, albumin/globulin ratio; ALB, albumin; ALP, alkaline phosphatase; CRP, C reactive protein; ESR, erythrocyte sedimentation rate; GLB, globulin; SACE, serum ACE; SC, serum calcium; sIL-2R, soluble interleukin 2 receptor; TB-AB: tuberculosis antibody; UA, blood uric acid; UC, urinary calcium; WCC, white cell count.

differences in body mass index, fatigue, Scadding stage distribution, serum calcium, serum ACE (SACE) and globulin levels. The airflow limitation was observed in 42.3% of patients with isolated pulmonary sarcoidosis and 42.2% of patients with pulmonary sarcoidosis and extrapulmonary involvement, with no statistical significance between these two groups (P=0.984). Similar result was seen in volume restriction between these two groups (7.1% vs 4.7%, P=0.198).

The differential characteristics of patients with or without deteriorating chest HRCT findings were shown in figure 2. Patients with pulmonary sarcoidosis with extrapulmonary involvement had more deteriorating HRCT findings than patients with isolated pulmonary

**Table 4** Pulmonary function parameters of subjects with pulmonary sarcoidosis, with and without extrapulmonary involvement

| Parameter | Isolated pulmonary sarcoidosis | Pulmonary sarcoidosis with extrapulmonary involvement | P value |
|---|---|---|---|
| VC, L | 3.04±0.75 | 3.09±0.82 | 0.327 |
| FVC, L | 3.03±0.75 | 3.07±0.81 | 0.309 |
| FVC % predicted | 97.5±15.6 | 96.8±17.2 | 0.305 |
| FEV1, L | 3.03±0.75 | 3.07±0.81 | 0.21 |
| FEV1 % predicted | 88.61±15.08 | 88.89±16.30 | 0.452 |
| FEV1/FVC | 78.62±7.74 | 78.94±8.52 | 0.072 |
| RV, L | 1.84±0.43 | 1.83±0.55 | 0.317 |
| TLC, L | 4.86±0.93 | 4.89±1.03 | 0.972 |
| RV/TLC | 38.51±7.66 | 37.73±9.16 | 0.137 |
| DLCO, mL/min/mm Hg | 21.1±4.8 | 20.6±5.2 | 0.487 |
| Airflow limitation, n (%) | 160 (42.3) | 109 (42.2) | 0.984 |
| Volume restriction, n (%) | 27 (7.1) | 12 (4.7) | 0.198 |

Data are presented as mean±SD or %.
DLCO, diffusion capacity for carbon monoxide of lung; FEV1, forced expiratory volume in 1 s; FVC, forced vital capacity; RV, residual volume; TLC, total lung capacity; VC, vital capacity.

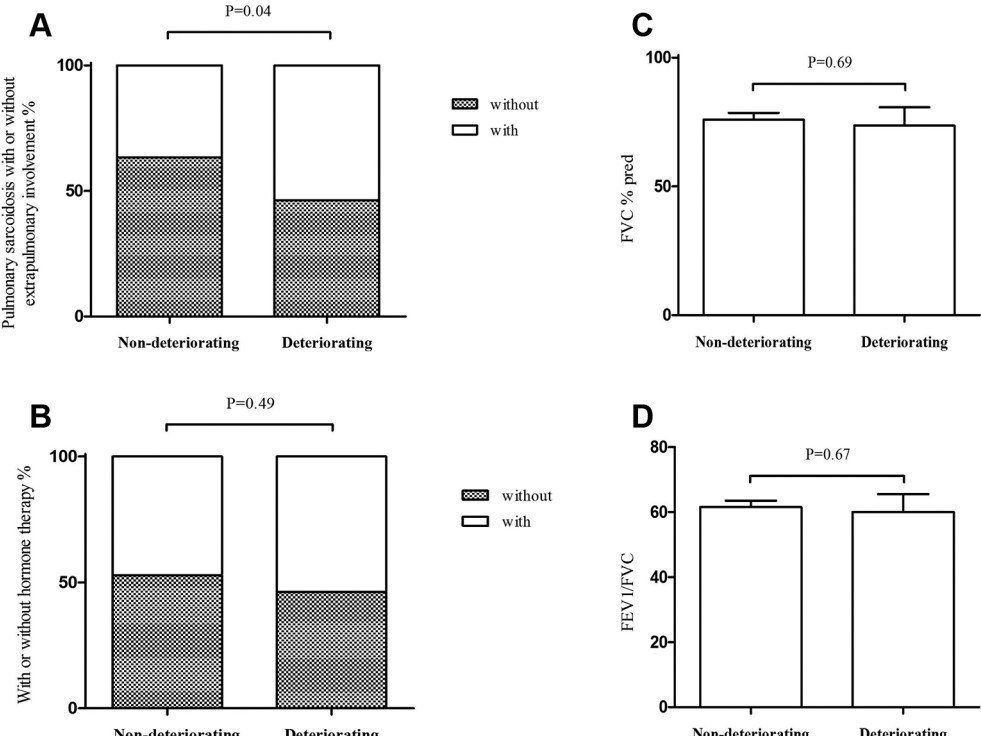

**Figure 2** (A) Pulmonary sarcoidosis with or without extrapulmonary involvement; (B) with or without corticosteroid therapy; (C) FVC % predicted; (D) FEV1/FVC ratio of patients who had deteriorating chest radiograph findings and patients who did not have deteriorating chest radiograph findings. FEV1, forced expiratory volume in 1 s; FVC, forced vital capacity; pred, predicted.

sarcoidosis (21 vs 18, P=0.04). Even though corticosteroid therapy, FVC% predicted and FEV1/FVC ratio were not statistically significant, we set these parameters as independent variables in the logistic regression model to investigate whether these factors had influence on prognosis of HRCT.

The OR values and 95% CIs of variables related to deteriorating HRCT findings in all patients were illustrated in table 5. However, the extrapulmonary involvement, corticosteroid therapy, FVC% predicted and FEV1/FVC ratio were irrelevant to the risk of deteriorating HRCT findings in these patients.

## DISCUSSION

The current study aimed to retrospectively compare the clinical characteristics of patients with pulmonary sarcoidosis with and without extrapulmonary involvement, as

**Table 5** Factors associated with deteriorating high-resolution CT findings in all subjects according to the logistic regression analysis

| Parameter | OR (95% CI) | P value |
|---|---|---|
| Extrapulmonary involvement | 0.516 (0.232 to 1.147) | 0.104 |
| Corticosteroid therapy | 0.941 (0.42 to 2.11) | 0.884 |
| FVC% predicted | 0.998 (0.978 to 1.108) | 0.843 |
| FEV1/FVC ratio | 0.998 (0.954 to 1.043) | 0.928 |

FEV1, forced expiratory volume in 1 s; FVC, forced vital capacity.

well as their prognosis on HRCT. So far, several studies conducted by different countries have been undertaken to investigate extrapulmonary sarcoidosis except for China. Since clinical features of sarcoidosis have varied among various populations around the world, our study would provide valuable information and, therefore, may help facilitate the prognosis and management for Chinese patients with sarcoidosis.

The prevalence of extrapulmonary sarcoidosis involvement varies among populations. Based on our study, we found that 40.6% patients with pulmonary sarcoidosis had extrapulmonary involvement, which was consistent with previous study in which 30%–50% of patients showed extrapulmonary disease localisations.[7] The most common sites found in our study were extrathoracic lymph nodes and skin, which is similar to studies from Turkey,[12 13] but differs from other researches. Skin lesions are divided into two categories: specific and non-specific. In this study, most of the skin lesions are specific lesions that demonstrate granulomatous inflammation on biopsy. The rest are non-specific lesions that show no granulomatous inflammation, such as erythema nodosum. ACCESS demonstrated that Caucasians more frequently had a sarcoidosis-related calcium metabolism disorder,[6] Finnish and Japanese patients were found to have higher rates of cardiac and eye involvement,[14] Puerto Ricans were common with lupus pernio skin lesions and Europeans had more erythema nodosum lesions.[1] Extrapulmonary sarcoidosis may be more frequent in females, illustrated by a health maintenance organisation database[15] and a

study from Beijing Tongren Hospital (Beijing, China).[16] However, we did not find the difference between gender in this study.

The most common symptom of mediastinal–pulmonary manifestations of sarcoidosis is persistent cough. Dyspnoea is rare in the early stages, but more frequent later on.[3] Another symptom is fatigue that occurs in up to 70% of patients, which can be measured and monitored with a validated fatigue assessment scale[17] which could predict quality of life.[18] Previous research showed that patients with pulmonary and extrapulmonary sarcoidosis were more fatigued and more dyspnoeic than those with pulmonary involvement alone, which demonstrated differences in the severity of symptoms.[19] In our study, a portion of patients had no symptoms, so we analysed the proportion of various complaints and found that patients with extrapulmonary involvement had higher proportion of fatigue than those with isolated pulmonary sarcoidosis (16.6% vs 8.3%, P=0.013). Neither cough nor dyspnoea was found to be different between the two groups in our study.

Of note, in our study we found patients with pulmonary sarcoidosis with extrapulmonary involvement had higher level of SACE than patients with isolated pulmonary sarcoidosis (79.0±46.9 vs 69.7±38.7 IU/L, P=0.016). ACE is a carboxypeptidase enzyme that converts angiotensin I into angiotensin II[20] which is produced in the epitheliod cell of the sarcoid granuloma.[21] SACE level, especially the amount of ACE activity, has been proposed as a biomarker that reflects the burden of granulomas.[22 23] Previous research also revealed that SACE level had correlated with extrapulmonary organs involvement and overall disease activity.[24 25] SACE is related to the various polymorphisms of the ACE gene, so different races may have significant differences in their SACE levels.[26] Another significant biomarker in sarcoidosis is soluble interleukin 2 receptor (sIL-2R). A previous study showed that sIL-2R was higher in patients with extrapulmonary sarcoidosis than those with isolated pulmonary sarcoidosis.[27] However, in our study, there was no difference in sIL-2R concentration (1151±842 vs 1163±798 U/mL in patients with and without extrapulmonary involvement, respectively, P=0.32). Further studies may be necessary to demonstrate the connection between serum biomarkers and differences in races and phenotypes.

Using chest radiograph or CT to view the internal structure is the most common imaging biomarker for evaluating sarcoidosis. The sarcoidosis granulomas can be detected with plain radiographic or CT radiographic techniques.[28] In our study, patients with isolated pulmonary sarcoidosis had higher proportion in stage I (n=128, P<0.05), while patients with sarcoidosis with extrapulmonary involvement had higher proportion in stage II (n=180, P<0.05). Even though Scadding criteria provided useful prognostic information, these stages do not necessarily progress or regress from one to another.[29] CT is useful in assessing disease extent[30]; HRCT may also provide prognostic information.[31 32] However, Zappala

*et al* found that change in radiographic extent was more applicable to routine monitoring in sarcoidosis than change in radiographic stage.[33] So we used deteriorating chest HRCT findings (an increase in extent compared with the previous HRCT results) as prognosis of sarcoidosis. FVC is the most common end point in pulmonary sarcoidosis trials[34]; significant airway obstruction is also associated with sarcoidosis via different mechanisms.[26] Corticosteroids are considered the drug for the treatment of almost all forms of sarcoidosis.[35] Even though these three factors were not statistically significant between patients with and without deteriorating HRCT findings, we still considered and included them as independent variables in the logistic regression model. However, we did not find relevant risk factors of deteriorating HRCT findings in these patients.

There are several limitations in this study. First, there is a risk of selection bias due to the retrospective design. Therefore, we conducted a cross-sectional and observational study which might reduce the risk of selection bias. Second, we conducted this study more from a pulmonary perspective. Since sarcoidosis is a multisystem disease, a more comprehensive study could be performed to investigate other parameters related to sarcoidosis. Third, to date, no reliable prognostic biomarkers have been identified. Based on the previous research,[33] we used deteriorating chest HRCT findings as a prognostic factor of sarcoidosis. Further studies are needed to demonstrate potential biomarkers that would predict the prognosis. Finally, the generalisability of our findings may be limited due to the number of single-centres that participated in our study; hence, our findings require additional validation, including data from other languages and other institutions.

In summary, our results suggest that extrapulmonary involvement is common in patients with pulmonary sarcoidosis, with the most common sites being extrathoracic lymph nodes and skin. Patients with sarcoidosis with extrapulmonary involvement have more complaint of fatigue, higher SACE levels and more deteriorating HRCT findings than those with isolated pulmonary sarcoidosis, to which clinicians should pay attention.

**Contributors** W-JC has full access to all of the data in the study and takes responsibility for the integrity of the data and the accuracy of the data analysis. C-WL, R-JT and D-FZ are joint first authors. W-JC and XX contributed to the study concept and design; R-JT and M-HL contributed to data collection and interpretation; C-WL and D-FZ performed the statistical analyses; finally all authors discussed data results and approved this submission.

**Funding** This research was funded by the National Key Research and Development Plan of Precision Medicine Project (Grant No. 2016YFC0905700).

**Competing interests** None declared.

**Patient consent** Obtained.

**Ethics approval** Ethics Committee of Shanghai Pulmonary Hospital.

**Provenance and peer review** Not commissioned; externally peer reviewed.

**Data sharing statement** No additional data are available.

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
