## [Reviewer comments · BMJ Open]

ARTICLE DETAILS

TITLE (PROVISIONAL)	Pulmonary sarcoidosis with and without extrapulmonary involvement : a cross-sectional and observational study in China
AUTHORS	li, chengwei; tao, rujia; zou, danfeng; li, manhui; xu, xin; cao, weijun

VERSION 1 – REVIEW

REVIEWER	Dr Simon DUBREY Hillingdon Hospital Middlesex, UB8 3NN United Kingdom
REVIEW RETURNED	20-Sep-2017

GENERAL COMMENTS	This study answers a very straightforward question about patients with 'pulmonary and extra-pulmonary sarcoid' or 'isolated pulmonary sarcoidosis' - in some ways I feel the text is almost too long to demonstrate the results on this very concise result. It does however tell us something of the 'Chinese experience' with this disease, which up to now was somewhat lacking in the literature. I have a couple of minor issues with the text itself. On page 6, lines 40-43 do not make sense - the English or grammar is wrong in this sentence. On page 6, line 51 - these results appear to be the wrong way around (having said one group has a higher value than the other, the numbers seem to show the opposite). The tables are clear and the images provide a balanced contrast to the text. The references all seem appropriate.
--

REVIEWER	Jan-Peter Smedema Maastricht University Medical Centre, The Netherlands Netcare Blaauwberg Hospital, South Africa
REVIEW RETURNED	14-Nov-2017

GENERAL COMMENTS	The authors do not include data on critical diagnostic tests like PET/CT, contrastedCMR, electrocardiography etc which are relevant to the diagnosis, the extent and management of extra-pulmonary sarcoidosis.
--

REVIEWER	Arda Kiani Tracheal Diseases Research Center, National Research Institute of
-----------------	---

	Tuberculosis and Lung Diseases (NRITLD), Shahid Beheshti University of Medical Sciences, Tehran, Iran
REVIEW RETURNED	26-Nov-2017

GENERAL COMMENTS	The manuscript is acceptable in the presented version and I find it as a valuable work.
---

REVIEWER	Helmut Popper Medical University Graz Austria
REVIEW RETURNED	01-Dec-2017

GENERAL COMMENTS	Drs. Li and colleagues report on pulmonary sarcoidosis in Chinese patients comparing patients with and without extrapulmonary involvement. They collected 636 cases, 378 with isolated lung, and 258 with extra pulmonary involvement. Skin and extra thoracic lymph nodes involvement was most frequently seen. There are some open questions which should be addressed:  1. Was there any rapid onset of symptoms such as Loeffgrens in their cases with skin involvement? 2. Interestingly there was a low incidence for heart, salivary glands, eyes/lacrimal glands, and also liver, spleen, involvements in their cases. This raises the question about how homogeneous the study population is? There are many ethnicities in China, which may present with different disease manifestations. Was the majority of the patients Han-Chinese? 3. As the authors claim to present an overview on Chinese patients, they need to have many cases from the major ethnicities in their database, otherwise it should be changed accordingly. 4. The authors should explain the type of skin lesions they have recorded 5. Other aspects should also be mentioned: Did the authors had cases of nodular sarcoidosis and also necrotizing sarcoid granulomatosis?
---

VERSION 1 – AUTHOR RESPONSE

Dear Prof. Hemali Bedi,

We thank you very much for giving us the opportunity to revise our manuscript. The comments of the reviewers are helpful and constructive. We have addressed the comments raised by the reviewers and revised the manuscript accordingly. Point-by-point responses to the reviewer's comments are listed below.

Thank you very much for your work!

Look forward to hearing good news from you soon.

With best wishes,

Wei-Jun Cao

Department of Respiratory Medicine

Shanghai Pulmonary Hospital

Tongji University School of Medicine

Shanghai, China

We would like to express our sincere thanks to the reviewers and the editor for the constructive and positive comments.

Responses to editor

1. Please work to improve the quality of the English throughout your manuscript. We encourage you to ask a native English speaking colleague to assist you.

Response : Thanks for the editor's suggestion. The manuscript was revised accordingly and was edited by a native English speaking professional.

2. Please revise your title to state the research question, study design, and setting (location). This is the preferred format for the journal.

Response : Thanks for the editor's suggestion. The title was revised as "pulmonary sarcoidosis with and without extrapulmonary involvement : a cross-sectional and observational study", which could state our research question, study design, and setting.

Responses to reviewers

Reviewer 1: Dr Simon DUBREY

1. On page 6, lines 40-43 do not make sense - the English or grammar is wrong in this sentence.

Response : Sorry for our unclear description. Changes have been made in the discussion part as below: "Previous research showed that patients with pulmonary and extrapulmonary sarcoidosis were more fatigued and more dyspneic than those with pulmonary involvement alone, which demonstrated differences in the severity of symptoms."

2. On page 6, line 51 - these results appear to be the wrong way around (having said one group has a higher value than the other, the numbers seem to show the opposite). The tables are clear and the images provide a balanced contrast to the text. The references all seem appropriate.

Response : We are sorry for the omission. Changes have been made as you suggested.

Reviewer 2: Prof. Jan-Peter Smedema

The authors do not include data on critical diagnostic tests like PET/CT, contrasted CMR, electrocardiography etc which are relevant to the diagnosis, the extent and management of extra-pulmonary sarcoidosis.

Response : We agree with the reviewer's comment. In this study organ involvement was determined and classified in each patient in accordance with criteria proposed in the ACCESS formulation [References 9], based upon clinical assessment and widely available tests, including the evolution of electrocardiography, positron emission tomography (PET) scans as well as magnetic resonance imaging (MRI) that can improve the detection of disease. Changes have been made in the method part as you suggested.

Reviewer 3: Prof. Arda Kiani

The manuscript is acceptable in the presented version and I find it as a valuable work.

Response : Thanks for the reviewer's suggestion.

Reviewer 4: Prof. Helmut Popper

1. Was there any rapid onset of symptoms such as Löfgrens in their cases with skin involvement?

Response : Thanks for the reviewer's suggestion. We are sorry that the symptoms of skin involvement are not included in this study, because we collected data from inpatients diagnosed with sarcoidosis at the Shanghai Pulmonary Hospital, and we conducted this study more from a pulmonary perspective, which is one limitation of this study. Since sarcoidosis is a multisystem disease, a more comprehensive study could be performed to investigate other parameters related to sarcoidosis.

2. Interestingly there was a low incidence for heart, salivary glands, eyes/lacrimal glands, and also liver, spleen, involvements in their cases. This raises the question about how homogeneous the study population is? There are many ethnicities in China, which may present with different disease manifestations. Was the majority of the patients Han-Chinese?

Response : Thanks for the reviewer's suggestion. We agree with the reviewer's comment that different ethnicities in China may present with different disease manifestations. In this study most of patients are Han nationality, and based on our study, we found both similar and different results from

other ethnicities all over the world. In the future, more prospective studies will be performed to pay attention to ethnicities in China.

3. As the authors claim to present an overview on Chinese patients, they need to have many cases from the major ethnicities in their database, otherwise it should be changed accordingly.

Response : Thanks for the reviewer's suggestion. We agree with the reviewer's comment that the cases cannot present an overview on Chinese patients. Changes have been made accordingly in the manuscript as you suggested.

4. The authors should explain the type of skin lesions they have recorded.

Response : Thanks for the reviewer's suggestion. Skin lesions are divided into two categories: specific and nonspecific. In this study most of skin lesions are specific lesions that demonstrate granulomatous inflammation on biopsy. The rest are nonspecific lesions that show no granulomatous inflammation, such as erythema nodosum. Changes have been made in the discussion part as you suggested.

5. Other aspects should also be mentioned: Did the authors had cases of nodular sarcoidosis and also necrotizing sarcoid granulomatosis?

Response : Thanks for the reviewer's suggestion. Previous study showed a striking overlap in the clinical, radiologic, and pathologic features of both nodular sarcoidosis and necrotizing sarcoid granulomatosis, supporting the conclusion that necrotizing sarcoid granulomatosis is a previously unrecognized manifestation of sarcoidosis and is essentially the same as nodular sarcoidosis. [Rosen Y. Four decades of necrotizing sarcoid granulomatosis: what do we know now? Arch Pathol Lab Med. 2015 Feb;139(2):252-62] In this study, diagnosis of sarcoidosis is based on standardized criteria[References 1], which detailed the features of sarcoidosis, including clinical and radiological presentation, and pathological findings (noncaseating epithelioid granulomas), as well as evidence of no alternative diseases. We also agree that concept of sarcoidosis should be expanded to recognize that there is a continuous spectrum of necrosis ranging from minimal to extensive.

VERSION 2 – REVIEW

REVIEWER	Helmut Popper Medical University, Pathology, Graz, Austria
REVIEW RETURNED	03-Jan-2018
GENERAL COMMENTS	Drs. Li and colleagues have responded to the critics adequately by omitting statements which were not correct and added missing information.

VERSION 2 – AUTHOR RESPONSE

Dear Editors and Reviewers:

Thank you for your letter and for the reviewers' comments concerning our manuscript entitled "Pulmonary sarcoidosis with and without extrapulmonary involvement : a cross-sectional and observational study" (ID: bmjopen-2017-018865.R1). Those comments are all valuable and very helpful for revising and improving our paper, as well as the important guiding significance to our researches. We have studied comments carefully and have made correction which we hope meet with approval. Revised portion are marked in red in the paper. The main corrections in the paper and the responds to the reviewer's comments are as flowing:

Reponses to editor

1. Please revise your title to include the location. This is the preferred format for the journal.

Response : Thanks for the editor's suggestion. The title was revised as "pulmonary sarcoidosis with and without extrapulmonary involvement : a cross-sectional and observational study in China", which included the location.

2. Please work to improve the quality of English throughout the manuscript, either with the help of a native speaking colleague or with the assistance of a professional copyediting agency.

Response : Thanks for the editor's suggestion. The manuscript was revised accordingly and was edited by a native English speaking professional.

3. Please add a description of the generalisability of the results to the discussion section, as per the requirements of the STROBE checklist.

Response : Thanks for the editor's suggestion. The generalisability has been added in the discussion part as below: "Finally, the generalisability of our findings may be limited due to the number of single-center that participated in our study, hence, our findings require additional validation, including data from other languages and other institutions."

Reponses to reviewers

Reviewer 4: Prof. Helmut Popper

1. Drs. Li and colleagues have responded to the critics adequately by omitting statements which were not correct and added missing information.

Response : Special thanks to you for your good comments.